# AUTOSPARSE: TOWARDS AUTOMATED SPARSE TRAINING

## ABSTRACT

Sparse training is emerging as a promising avenue for reducing the computational cost of training neural networks. Several recent studies have proposed pruning methods using learnable thresholds to efficiently explore the non-uniform distribution of sparsity inherent within the models. In this paper, we propose Gradient Annealing (GA), a gradient driven approach where gradients of pruned out weights are scaled down in a non-linear manner. GA provides an elegant trade-off between sparsity and accuracy without the need for additional sparsity-inducing regularization. We integrated GA with the latest learnable threshold based pruning methods to create an automated sparse training algorithm called AutoSparse. Our algorithm achieves state-of-the-art accuracy with $80\%$ sparsity for ResNet50 and $75\%$ sparsity for MobileNetV1 on Imagenet-1K. AutoSparse also results in $7\times$ reduction in inference FLOPS and $> 2\times$ reduction in training FLOPS for ResNet50 on ImageNet at $80\%$ sparsity. Finally, GA generalizes well to fixed-budget (Top-K, $80\%$) sparse training methods, improving the accuracy of ResNet50 on Imagenet-1K, to outperform TopKAST+PP by $0.3\%$. MEST (SotA method for fixed-budget sparsity) achieves comparable accuracy as AutoSparse at $80\%$ sparsity, however, using $20\%$ more training FLOPS and $45\%$ more inference FLOPS.

## 1 INTRODUCTION

Deep learning models have emerged as the preferred solution for many important problems in the domains of computer visionHe et al. (2016); Dosovitskiy et al. (2021), language modeling Brown et al. (2020), recommender systemsNaumov et al. (2019) and reinforcement learningSilver et al. (2017). Models have grown larger and more complex over the years, as they are applied to increasingly difficult problems on ever-growing datasets. In addition, DNN models are designed to operate in overparameterized regime Arora et al. (2018); Belkin et al. (2019); Ardalani et al. (2019) to facilitate easier optimization using gradient descent based methods. As a consequence, computational costs of performing training and inference tasks on state-of-the-art models has been growing at an exponential rateAmodei & Hernandez. The excess model capacity also makes DNNs more resilient to noise during training – reduced precision training methodsMicikevicius et al. (2017); Wang et al. (2018); Sun et al. (2019) have successfully exploited this aspect of DNNs to speed up training and inference tasks significantly. Today, state-of-the-art training hardware consists of significantly more reduced-precision FLOPs compared to traditional FP32 computations.

Sparsity is another avenue for improving the compute efficiency by exploiting excess model capacity, thereby reducing the number of FLOPs needed for each iteration. Several studies have shown that while overparameterized model helps to ease the training, only a fraction of that model capacity is needed to perform inference Li et al. (2020); Han et al. (2015); Li et al. (2020); Narang et al. (2017); Ström (1997); Gale et al. (2019). A wide array of studies have also proposed methods to prune dense networks to produce sparse models for inference (dense-to-sparse) Molchanov et al. (2017); Zhu & Gupta (2017); Frankle & Carbin (2019); Renda et al. (2020). More recently, there is a growing interest in sparse-to-sparse Frankle & Carbin (2019); Mostafa & Wang (2019); Bellec et al. (2018); Evci et al. (2021); Lee (2021); Dettmers & Zettlemoyer (2019); Jayakumar et al. (2021); Zhang et al. (2022); Schwarz et al. (2021); Yuan et al. (2021) methods training models with end-to-end sparsity to reduce the computational costs of training. This paper presents techniques to improve and generalize sparse training methods for easy integration into different training workflows.

Sparse training methods can be broadly divided into two categories – a) **deterministic pruning** methods, that initialize the model with a desired fixed sparsity budget at each layer and enforce it throughout the training cycle, b) **learnable threshold pruning** methods attempt to discover the sparsity distribution within the model by learning layer-wise threshold parameters. While these latter methods can aim for a desired sparsity level by selecting appropriate combination of initialization and hyper-parameters, the final model sparsity may not be exactly what was desired – hence they are non-deterministic. Please refer to Hoefler et al. (2021) for detailed categorization and discussion on various sparsification methods.

**Deterministic Pruning:** Deterministic pruning methods expect a prior knowledge of how much sparsity can be extracted out of any given model. This sparsity budget is often determined by trial-and-error or extrapolated from previously published studies. Once the sparsity budget is determined, a choice must be made between a uniform or a non-uniform distribution of sparsity across the layers. Majority of the methods in this category Frankle & Carbin (2019); Bellec et al. (2018); Evci et al. (2021); Lee (2021); Jayakumar et al. (2021); Zhang et al. (2022); Schwarz et al. (2021); Zhou et al. (2021); Yuan et al. (2021) opt for uniform sparse distribution because it requires fewer hyperparameters – a subset of these methods Jayakumar et al. (2021); Zhang et al. (2022); Schwarz et al. (2021) maintain first and last layers in dense while the sparsity budget is uniformly distributed across rest of the layers. Fewer methods in this category Mostafa & Wang (2019); Dettmers & Zettlemoyer (2019) use non-uniform distribution across layers using dynamic weight reallocation heuristics. Non-uniform distribution allows more degrees of freedom to explore ways to improve accuracy at any given sparsity budget. The best performing methods in this category are the ones that skip pruning the first and last layers.

## 1.1 LEARNABLE THRESHOLD PRUNING

Learnable threshold pruning methods offer a two-fold advantage over deterministic pruning methods, **1) computationally efficient**, as the overhead of computation (e.g. choosing Top-K largest) used for deriving threshold values at each layer is eliminated, **2) learns the non-uniform sparsity distribution** inherent within the model automatically, producing a *more FLOPs-efficient sparse model for inference*. For example, 80% sparse ResNet50 produced via fixed-budget method MEST requires 50% more FLOPS than learned sparsity method AutoSparse (discussed later) as compute profile for various layers is non-uniform (Figure 1a).

Current state-of-the-art methods in this space Liu et al. (2020); Kusupati et al. (2020) rely on $L2$ regularization based approaches to guide threshold updates and penalize pruned weights. Dynamic Sparse Training Liu et al. (2020) is sparse training method that starts with threshold values initialized to 'zero' and pushes the small thresholds up using an exponential loss function added to $L2$ regularization term. Soft Threshold Reparameterization Kusupati et al. (2020) initializes threshold parameters with large negative values and controls the induction of sparsity using different weight decay ($\lambda$) values for achieving different sparsity levels using $L2$ regularization. Sparsity vs accuracy trade-off is a challenge and is discussed below.

**Exploring Accuracy Vs. Sparsity Trade-Off:**

Deterministic pruning methods, not withstanding their limitations, offer a consistent level of sparsity throughout the training which in turn can lead to predictable performance improvements. For learnable threshold methods, the performance can be measured in the form of reduction in training FLOPs measured across the entire duration of the training. In order to meet this goal, learnable threshold methods must have the ability to induce sparsity early in the training and provide algorithmic means to explore the trade-off to increase model sparsity while also reducing accuracy loss. Empirical studies on aforementioned methodsLiu et al. (2020); Kusupati et al. (2020) have indicated that $L2$ regularization based approach offers at best, a weak trade-off of better accuracy at the expense of lower average sparsity. We also found these methods to be susceptible to *runaway sparsity* (e.g. hit 100% model sparsity), if higher levels of sparsity were induced right from the start of the training. To mitigate this problem, DST Liu et al. (2020) implements hard upper limit checks (e.g. 99%) on sparsity to trigger a reset of the offending threshold and all the associated weights to prevent loss of accuracy. Similarly, STR Kusupati et al. (2020) methods uses a combination of small initial threshold values with an appropriate $\lambda$ to delay the induction of sparsity until later in the training cycle (e.g. 30 epochs) to control unfettered growth of sparsity.

Motivated by the above challenges, we propose **Gradient Annealing (GA)** method, to address the aforementioned issues related to training sparse models. When compared existing methods, **GA** offers greater flexibility to explore the trade-off between model sparsity and accuracy, and provides greater stability by preventing divergence due to runaway sparsity. We also propose a unified training algorithm called **AutoSparse**, combining best of learnable threshold methods Kusupati et al. (2020) with **GA** that attempts to pave the path towards full automation of sparse-to-sparse training. Additionally, we also demonstrated that when coupled with deterministic pruning methods (**TopKAST**), **Gradient Annealing** can extract better accuracy at a fixed ($80\%$) sparsity budget. The key contributions of this work are as follows:

• We present a novel *Gradient Annealing* (GA) method (+ hyper-parameter $\alpha$) which is a generalized and more accurate gradient approximator than STE (Bengio et al. (2013)) and ReLU (e.g. STR Kusupati et al. (2020)). For training end-to-end sparse models, GA provides greater flexibility to explore sparsity and accuracy trade-off than other methods.

• We propose *AutoSparse*, a unified algorithm that combines learnable threshold methods with *Gradient Annealing* to create an automated framework for end-to-end sparse training. Our algorithm outperformed state-of-the-art methods by achieving better accuracy while maintaining consistently high sparsity throughout the training for ResNet50 and MobileNetV1 on Imagenet-1K dataset.

• AutoSparse achieves best FLOPs efficiency for inference and training among methods with comparable sparsity and accuracy.

• We demonstrate the efficacy of *Gradient Annealing* as a general learning technique independent of *AutoSparse* by applying it to *TopKAST* method to improve the Top-1 accuracy of ResNet50 by $0.3\%$ on ImageNet-1K dataset using same fixed sparsity budget of $80\%$.

## 2 GRADIENT ANNEALING (GA)

A typical pruning step of deep networks involves masking out weights that are below some threshold $T$. This sparse representation of weights benefits from sparse computation in forward pass and in computation of gradients of inputs. We propose the following pruning step, where $w$ is a weight and $T$ is a threshold that can be deterministic (e.g., TopK magnitude) or learnable:

$$(\text{sparse}) \quad \tilde{w} = \text{sign}(w) \cdot h_\alpha(|w| - T)$$

$$\text{Forward pass} \quad h_\alpha(x) = \begin{cases} x, & x > 0 \\ 0, & x \leq 0 \end{cases} \qquad (\text{Proxy}) \text{ Gradient} \quad \frac{\partial h_\alpha(x)}{\partial x} = \begin{cases} 1, & x > 0 \\ \alpha, & x \leq 0 \end{cases} \tag{1}$$

where $0 \leq \alpha \leq 1$. $\tilde{w}$ is 0 if $|w|$ is below threshold $T$. Magnitude-based pruning is a greedy, temporal view of parameter importance. However, some of the pruned-out weights (in early training epochs) might turn out to be important in later epochs when a more accurate sparse pattern emerges. For this, $h_\alpha(\cdot)$ in eqn (1) allows the loss gradient to flow to masked weights in order to avoid permanently pruning out some important weights. The proposed gradient approximation is inspired by the Straight Through Estimator (STE) Bengio et al. (2013) which replaces zero gradients of discrete sub-differentiable functions by proxy gradients in back-propagation. Furthermore, we decay this $\alpha$ as the training progresses. We call this technique the *Gradient Annealing*.

We anneal $\alpha$ at the beginning of every epoch and keep this value fixed for all the iterations in that epoch. We want to decay $\alpha$ slowly in early epochs and then decay it steadily. For this, we compare several choices for decaying $\alpha$: fixed scale (no decay), linear decay, cosine decay (same as learning rate (2)), sigmoid decay (defined in (3)) and Sigmoid-Cosine decay (defined in (4)). For sigmoid decay in (3), $L_0 = -6$ and $L_1 = 6$. For total epochs $T$, scale for $\alpha$ in epoch $i$ is

$$\begin{aligned} \text{Cosine-Decay}(i, T) \qquad c_i &= (1 + \text{cosine}(\pi \cdot i/T))/2 & (2) \\ \text{Sigmoid-Decay}(i, T) \qquad s_i &= 1 - \text{sigmoid}(L_0 + (L_1 - L_0) \cdot i/T) & (3) \\ \text{Sigmoid-Cosine-Decay}(i, T) \qquad &= \max\{s_i, c_i\} & (4) \end{aligned}$$

Figure 1 shows the effect of various linear and non-linear annealing of $\alpha$ on dynamic sparsity. Fixed scale with no decay (STE) does not give us a good control of dynamic sparsity. Linear decay is better than this but suffers from drop in sparsity towards the end of training. Non-linear decays in eqn (2, 3,

4) provide much superior trade-off between sparsity and accuracy. While eqn (2) and eqn (4) show very similar behavior, sharp drop of eqn (3) towards the end of training push up the sparsity a bit (incurring little more drop in accuracy). These plots are consistent with our analysis of convergence of GA in eqn (5). Annealing schedule of $\alpha$ closely follows learning rate decay schedule.

**Analysis of Gradient Annealing**

Here we analyze the effect of the transformation $h_\alpha(\cdot)$ on the convergence of the learning process using a simplified example as follows. Let $v = |w| - T$, $u = h_\alpha(v)$, optimal weights be $w^*$ and optimal threshold be $T^*$, i.e., $v^* = |w^*| - T^*$. Let us consider the loss function as

$$\min_v \mathcal{L}(v) = 0.5 \cdot (h_\alpha(v) - v^*)^2$$

and let $\partial h_\alpha(v)$ denote the gradient $\frac{\partial h_\alpha(v)}{\partial v}$. We consider the following cases for loss gradient for $v$.

$$\frac{\partial \mathcal{L}}{\partial v} = \partial h_\alpha(v)(h_\alpha(v) - v^*) = \begin{cases} \partial h_\alpha(v) \cdot 0 = 0 & \text{if} \quad h_\alpha(v) = v^* \\ \partial h_\alpha(v) \cdot (v - v^*) = 1 \cdot (v - v^*) & \text{if} \quad v > 0 \text{ and } v^* > 0 \\ \partial h_\alpha(v) \cdot (v + |v^*|) = 1 \cdot (v + |v^*|) & \text{if} \quad v > 0 \text{ and } v^* \leq 0 \\ \partial h_\alpha(v) \cdot (-v^*) = \alpha \cdot (-v^*) & \text{if} \quad v \leq 0 \text{ and } v^* > 0 \\ \partial h_\alpha(v) \cdot (|v^*|) = \alpha \cdot (|v^*|) & \text{if} \quad v \leq 0 \text{ and } v^* \leq 0 \end{cases} \quad (5)$$

Correct proxy gradients for $h_\alpha(\cdot)$ should move $v$ towards $v^*$ during pruning (e.g., opposite direction of gradient for gradient descent) and stabilize it at its optima (no more updates). Therefore, $\partial h_\alpha(v) > 0$ should be satisfied for better convergence of $v$ to $v^*$. Our $h_\alpha(\cdot)$ satisfies this condition for $\alpha > 0$. Furthermore, for $v > 0$, $v$ gets updated proportional to $v - v^*$, i.e., how far $v$ is from $v^*$. As training progresses and $v$ gets closer to $v^*$, $v$ receives gradually smaller gradients to finally converge to $v^*$. However, for $v \leq 0$, $v$ receives gradient proportional to magnitude of $\alpha \cdot v^*$, irrespective of how close $v$ is to $v^*$. Also, note that we benefit from sparse compute when $v \leq 0$.

We set initial $T$ high in order to achieve sparsity from early epochs. However, this likely leads to a large number of weights following condition 4 in eqn (5). Fixed, large $\alpha$ (close to 1) makes large correction to $v$ and moves it to $v^*$ quickly. Consequently, $v$ moves from condition 4 to condition 2, losing out the benefit of sparse compute. A lower $\alpha$ 'delays' this transition and enjoys the benefits of sparsity. *This is why we choose $\alpha < 1$ rather than identity STE as proxy gradient* (unlike Tang et al. (2022)). However, as training progresses, more and more weights move from condition 4 to condition 2 leading to a drop in sparsity. This behavior is undesirable to reach a target sparsity at the end of training. In order to overcome this, we propose to decay $\alpha$ with training epochs such that we enjoy the benefits of sparse compute while $v$ being close to $v^*$. That is, GA provides a more controlled and stable trade-off between sparsity and accuracy throughout the training.

Note that, GA is applicable when we compute loss gradients for a superset of active (non-zero) weights that take part in forward sparse computation using gradient descend. For an iteration $t$, let the forward sparsity be $S$. If $\alpha = 0$, then we need to calculate gradient for only those non-zero weights as other weights would not receive gradients due to ReLU STE. In order to benefit from such computational reduction, we can set $\alpha = 0$ after several epochs of $\alpha$ annealing.

## 3 AUTOSPARSE : SPARSE TRAINING WITH GRADIENT ANNEALING

AutoSparse is the sparse training algorithm that combines the best of learnable threshold pruning techniques Kusupati et al. (2020); Liu et al. (2020) with Gradient Annealing (GA). AutoSparse meets the requirements necessary for efficient training of sparse neural networks, as outlined in Section 1.1.

• **Learnable Pruning Thresholds** : Eliminate threshold computation, reduce sparsification overhead compared to deterministic pruning methodsJayakumar et al. (2021). Learn the non-uniform distribution of sparsity across the layers.

• **Sparse Model Discovery** : Discover an elegant trade-off between model accuracy vs. level of sparsity by applying Gradient Annealing method (as shown in Figure 1). Produce a sparse model at the end of the training with desired sparsity level guided by the hyper-parameter $\alpha$.

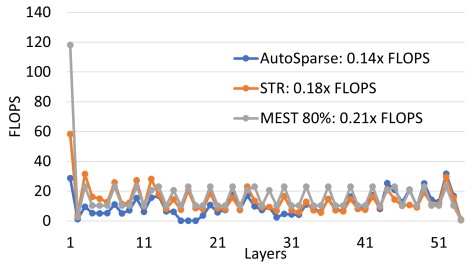

(a) Inference FLOPS for 80% sparse ResNet50 produced by fixed-budget and learned sparsity methods

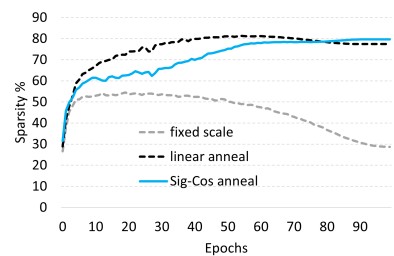

(b) Sparsity achieved for various gradient annealing

Figure 1: Sparse ResNet50 training on ImageNet

• **Accelerate Training/Inference** : Reduce training FLOPs by training with sparse weights from scratch, maintaining high levels of sparsity throughout the training, and using sparsity in both forward and backward pass. Produce FLOPS-efficient sparse model for inference.

Previous proposals using leanable threshold methods such as Liu et al. (2020) and Kusupati et al. (2020) address the first criterion but do not effectively deal with accuracy vs sparsity trade-off as they are outperformed by our method. These methods also do not accelerate training by reducing FLOPS as effectively as our method. Deterministic pruning methods such as Jayakumar et al. (2021); Evci et al. (2021); Yuan et al. (2021) address the third criterion of accelerating training, but incur higher sparsfication overheads for computing threshold values and cannot automatically discover the non-linear distribution of sparsity. This leads to sub-optimal sparse models for inference (Figure 1a).

**Formulation:** Let $\mathcal{D} := \{(\mathbf{x_i} \in \mathbb{R}^{\mathbf{d}}, y_i \in \mathbb{R})\}$ be the observed data, $\mathcal{W}$ be the learnable network parameters, $\mathcal{L}$ be a loss function. For an $L$-layer DNN, $\mathcal{W}$ is divided into layer-wise trainable parameter tensors, $[\mathbf{W}_\ell]_{\ell=1}^L$. As various layers can have widely different number of parameters and also unequal sensitivity to parameter alteration, we use one trainable pruning parameter, $s_\ell$ for each layer $\ell$, i.e., $\mathbf{s} = [s_1, ..., s_L]$ is the vector of trainable pruning parameter. Let $g : \mathbb{R} \to \mathbb{R}$ be applied element-wise. For layer $\ell$, $T_\ell = g(s_\ell)$ is the pruning threshold for $\mathbf{W}_\ell$. We seek to optimize:

$$\min_{\mathcal{W}, \mathbf{s}} \mathcal{L}(\mathcal{S}_{h_\alpha, g}(\mathcal{W}, \mathbf{s}), \mathcal{D}) \tag{6}$$

where, function $\mathcal{S}_{h_\alpha, g}$, parameterized by $h_\alpha \in \mathbb{R} \to \mathbb{R}$ that is applied element-wise.

$$\hat{\mathbf{W}}_\ell = \mathcal{S}_{h_\alpha, g}(\mathbf{W}_\ell, s_\ell) = \text{sign}(\mathbf{W}_\ell) \cdot h_\alpha(|\mathbf{W}_\ell| - g(s_\ell)) \tag{7}$$

Gradient annealing is applied via $h_\alpha(\cdot)$ as discussed earlier.

**Sparse Forward Pass:** At iteration $t$, for layer $\ell$, sparse weights are $\hat{\mathbf{W}}_\ell^{(t)} = \mathcal{S}_{h_\alpha, g}(\mathbf{W}_\ell^{(t)}, s_\ell^{(t)})$ as defined in eqn (7). Let the non-zero (active) set of weights $\mathcal{A}_\ell^{(t)} = \{i : \hat{\mathbf{W}}_{\ell,i}^{(t)} > 0\}$. For simplicity, let us drop the subscript notations. $\hat{\mathbf{W}}$ is used in the forward pass as $\mathbf{Y} = \mathbf{X} \otimes \hat{\mathbf{W}}$ where $\otimes$ is tensor MAC operation, e.g., convolution. Let $A$ denote the fraction of non-zero elements of $\mathbf{W}$ belonging to $\mathcal{A}$, i.e, out forward sparsity is $1 - A$. $\hat{\mathbf{W}}$ is also used in computation of input gradient during backward pass. Every iteration, we update $\mathbf{W}$ and construct $\hat{\mathbf{W}}$ from updated $\mathbf{W}$ and learned threshold.

**Sparse Compute of Input Gradient:** For an iteration and a layer (we drop the subscripts $t$, $\ell$ for simplicity), output of sparse operations in forward pass need not be sparse, i.e., $\mathbf{Y} = \mathbf{X} \otimes \hat{\mathbf{W}}$ is typically dense. Consequently, gradient of output $\nabla_Y$ is also dense. We compute gradient of input $\nabla_X$ as $\hat{\mathbf{W}} \otimes \nabla_Y$. Computation of $\nabla_X$ is sparse due to sparsity in $\hat{\mathbf{W}}$.

**Sparse Weight Gradient:** Gradient of weights $\nabla_W$ is computed as $\mathbf{X} \otimes \nabla_Y$. Note that, for forward sparsity S, $\alpha = 0$ implies weight gradient sparsity S as no gradient flows to pruned weights. We can have a hyperparameter that specifies at which epoch we set $\alpha = 0$, to enjoy benefits of sparse weight gradients. However, we need to keep $\alpha \neq 0$ for several epochs to achieve high accuracy results, losing the benefits of sparse weight gradient. In order to overcome this, we propose the following for the epochs when $\alpha \neq 0$.

We can make sparse $\nabla_W$ if we compute loss gradient using a subset $\mathcal{B}$ of $\mathbf{W}$.

$$\mathcal{B} = \{i : \mathbf{W}_i \in \text{TopK}(\mathbf{W}, B)\}, \quad B \geq A \tag{8}$$

where $\mathcal{B}$ is a superset of $\mathcal{A}$ and $\text{TopK}(\mathbf{W},k)$ picks indices of $k$ largest magnitude elements from $\mathbf{W}$. This constitutes our weight gradient sparsity $1 - B$.

We apply gradient annealing on set $\mathcal{B} \setminus \mathcal{A}$, i.e., gradients of weights in $\mathcal{B} \setminus \mathcal{A}$ are decayed using $\alpha$. Note that, $\alpha = 0$ implies $\mathcal{B} = \mathcal{A}$. Gradient computation for parameters $\mathbf{W}$ and $s$ are in Appendix.

## 4 RELATED WORK

**Learnable Threshold Methods**

STR Kusupati et al. (2020) is a state-of-the-art method that learns pruning thresholds along with weights. STR prunes weights that are below the threshold using ReLU as a mask function. Setting $\alpha = 0$ in (7) reproduces the formulation of STR (however, our method learns completely different distribution than that of STR). This $\alpha = 0$ implies identical forward and backward sparsity which are computationally beneficial. However, this prevents gradients from flowing to pruned out weights resulting in a sub-optimal trade-off between sparsity and accuracy. This forces STR to run fully dense training for many epochs (implemented by initializing threshold parameter $s$ to a very large negative value, e.g., -3200 so that sigmoid($s$) $\approx 0$, and weight decay on $s$ takes many epochs to make sigmoid($s$) large enough to induce sparsity. Note that, extracting sparsity from early epochs using STR results in run-away sparsity. Using gradient annealing technique, we are able to extract sparsity from early epochs to maintain high average sparsity throughout the training, yet achieve SOTA accuracy.

DST Liu et al. (2020) is also a sparse training algorithm based on learned weights and thresholds, where a binary weight mask is created by suppressing the weights that are below the threshold (using step function). they impose exponential decay of thresholds as regularizer. A hyperparameter controls the amount of regularization, leading to a balance between sparsity and accuracy. In order to reduce the sensitivity of the hyperparameter, they manually reset the sparsity if it exceeds some predefined limit (thus occasionally falling into dense training regime). Further, they approximated the gradient of pruning step function by linearly scaling up the gradient of the elements that are close to the pruning threshold, and scaling down (fixed scale) the gradient of other elements. This helps some of the pruned out elements to receive loss gradient.

SCL Tang et al. (2022) learns both network weights as well as mask parameters during training. These masks are binarized during forward pass to represent network sparsity. This mask information, along with a decaying connectivity hyperparameter are used as a sparsity-inducing regularizer in the objective function. During training, the mask is learned as dense parameter tensor. This increases the effective model size during training, which might create overhead moving parameters from memory. LTP Azarian et al. (2021) learns the pruning thresholds using soft pruning and soft $L_0$ regularization where sigmoid is applied on transformed weights and sparsity is controlled by a hyper-parameter. Savarese et al. (2021) used sigmoidal soft-threshold function as a sparsity-inducing regularization.

**Deterministic Pruning Methods**

GMP Zhu & Gupta (2017) suggests to gradually increase the number of removed weights until desired sparsity is reached. RigL Evci et al. (2021) proposes to prune out a fraction of weights, and activates/revives new ones iteratively using infrequent full gradient calculations. Such restriction of gradient flow causes more accuracy drop. TopKAST Jayakumar et al. (2021) always prunes Top-K weights, but updates a superset of active weights based on gradients so that the pruned out weights can be revived. Moreover, they penalize the non-active weights by a factor inversely proportional to density. This way, the active set of weights becomes stable with epochs as small fraction of pruned out weights become active again. ITOP Liu et al. (2021) maintains a fixed amount of sparsity throughout training. They use gradient based weight regrowth used in RigL and SNFS Dettmers & Zettlemoyer (2019). Applying ITOP in RigL grow stage, i.e., exploring new weights based on (full) gradient information in Rigl grow stage, RigL+ITOP achieves higher accuracy. PowerPropagation Schwarz et al. (2021) is a technique to transform weight as $w = v|v|^{\alpha-1}$, s.t. it creates a heavy-tailed distribution of trained weights. As an effect, they observe that weights initialized close to 0 are likely to be pruned out, and weights are less likely to change sign. PP, when applied on TopKAST, i.e.,

TopKAST + PP, improves the accuracy of TopKAST. OptG Zhang et al. (2022) learns both weights and a pruning supermask in a gradient driven manner. They argued in favour of letting gradient flow to pruned weights so that it solves the 'independence paradox' problem that prevents from achieving high-accuracy sparse results. However, they achieve a given sparsity budget by increasing sparsity according to some sigmoid schedule (sparsity is extracted only after 40 epochs). This suffers from larger training FLOPs count. MEST (Yuan et al. (2021)) always maintains fixed sparsity in forward and backward pass by computing gradients of survived weights only. For better exploration, they remove some of the least important weights (ranked proportional to magnitude plus the gradient magnitude) and introduce same number of random 'zero' weights to maintain the sparsity budget. Similar to RigL, they also need to train a lot longer (250 epochs vs 500 epochs in RigL). Gradmax (Evci et al. (2022)) proposed to grow network by adding more weights gradually with training epochs in order to reduce overall training FLOPS. SWAT (Raihan & Aamodt (2020)) sparsifies both weights and activations by keeping TopK magnitudes in order to further reduce the FLOPS.

## 5 EXPERIMENTS

### 5.1 EXPERIMENTAL SET UP

**Vision Models:** ImageNet-1K (Deng et al. (2009)) is a widely used large-scale image classification dataset with 1K classes. We show sparse training results on ImageNet-1K for two popular CNN architectures: ResNet50 He et al. (2016) and and MobileNetV1 Howard et al. (2017), to demonstrate the generalizability of our method. For AutoSparse training, we use SGD as the optimizer, momentum $0.875$, learning rate (max) $0.256$ using a cosine annealing with warm up of 5 epochs. We run all the experiments for 100 epochs using a batch size 256. We use weight decay $\lambda = 0.000030517578125$ (picked from STR Kusupati et al. (2020)), label smoothing $0.1$, $s_0 = -5$. We presented our results only using Sigmoid-Cosine decay of $\alpha$ (defined in eqn (4)).

**Language Models:** We choose Transformer models Vaswani et al. (2017) for language translation on WMT14 English-German data. We have 6 encoder and 6 decoder layers with standard hyperparameter setting: optimizer is ADAM with betas $(0.9, 0.997)$, token size $10240$, warm up $4000$, learning rate $0.000846$ that follows inverse square root decay. We apply AutoSparse by introducing a learnable threshold for each linear layer, and we initialize them as $s = -7.0$. Also, initial $\alpha = 0.4$ is annealed according to exponential decay as follows. For epoch $t$ and $\beta > 0$ (we use $\beta = 1$):

$$\text{Exponential-Decay}\,(t; \beta) = e^{(-\beta \cdot t)} \tag{9}$$

We keep the first and last layers of transformer dense and apply AutoSparse to train it for $44$ epochs.

We repeat the experiments several times with different random seeds and report the average numbers.

**Notation:** We define the following notations that are used in the tables. 'Base': Dense baseline accuracy, 'Top1(S)': Top-1 accuracy for sparse models, 'Drop%': relative drop in accuracy for sparse models from Base, '$S\%$': percentage of model sparsity, 'Train F': fraction of training FLOPs comparing to baseline FLOPs, 'Test F': fraction of inference FLOPs comparing to bsaeline FLOPs, 'Back $S\%$': explicitly set sparsity in weight gradients ($B$ in eqn (8)), 'BLEU(S)': BLEU score for sparse models. Smaller values of 'Train F' and 'Test F' suggests larger reduction in computation.

### 5.2 EFFICACY OF GRADIENT ANNEALING

**Comparison with Other Learnable Threshold Methods:**

We compare AutoSparse results with STR Kusupati et al. (2020) and DST Liu et al. (2020). Lack of gradient flow to pruned out elements prevents STR to achieve the optimal sparsity-accuracy trade-off. For example, they need dense training for many epochs in order to achieve high accuracy results, losing out the benefits of sparse training. Our gradient annealing overcomes such problems and achieves much superior sparsity-accuracy trade-off. In Table 1, we achieve better accuracy than STR for ResNet50 at various sparsity levels $80\%$, $90\%$, and $95\%$, while gaining in FLOPs reduction for both training and inference. Also, Table 3 shows our superior accuracy for MobileNetV1. Similarly, our method achieves higher accuracy than DST for both $80\%$ and $90\%$ sparsity budget for ResNet50 (Table 1). However, DST uses separate sparsity-inducing regularizer, whereas our gradient annealing itself does the trick. SCL Tang et al. (2022) is another dynamic sparse method that learns both

| Method | Base | Top1(S) | Drop% | S% | Train F | Test F | comment |
|---|---|---|---|---|---|---|---|
| RigL | 76.8 | 74.6 | 2.86 | 80 | 0.33× | 0.22× | fixed sparsity budget |
| *SWAT-U | 76.8 | 75.2 | 2.08 | 80 | 0.24× | 0.22× | TopK weight, act |
| GMP | 77.01 | 75.6 | 1.83 | 80 | – | 0.2× | fixed sparsity budget |
| TopKAST* | 76.8 | 75.7 | 0.94 | 80 | 0.48× | 0.22× | fixed sparsity budget |
| TopKAST*+PP | 76.8 | 76.24 | 0.73 | 80 | 0.48× | 0.22× | fixed sparsity budget |
| TopKAST*+**GA** | 76.8 | 76.47 | 0.43 | 80 | 0.48× | 0.22× | fixed sparsity budget |
| $MEST_{1.7\times}$+EM | 76.9 | 76.71 | 0.25 | 80 | 0.57× | 0.21× | fixed sparsity budget |
| DST | 74.95 | 74.02 | 1.24 | 80.4 | – | 0.15× | learnable sparsity |
| STR | 77.01 | 76.19 | 1.06 | 79.55 | 0.54× | 0.18× | learnable sparsity |
| **AutoSparse** | 77.01 | 76.77 | 0.31 | 79.67 | 0.46× | 0.14× | $\alpha_0$=.75,$\alpha$=0@epoch90 |
| **AutoSparse** | 77.01 | 76.59 | 0.55 | 80.78 | 0.36× | 0.14× | $\alpha_0$=.8,$\alpha$=0@epoch70 |
| $MEST_{1.7\times}$+EM | 76.9 | 75.91 | 1.29 | 90 | 0.25× | 0.11× | fixed sparsity budget |
| OptG | 77.01 | 74.28 | 3.55 | 90 | – | – | |
| DST | 74.95 | 72.78 | 2.9 | 90.13 | – | 0.087× | learnable sparsity |
| STR | 77.01 | 74.31 | 3.51 | 90.23 | 0.38× | 0.083× | learnable sparsity |
| **AutoSparse** | 77.01 | 75.9 | 1.44 | 85.1 | 0.37× | 0.096× | $\alpha_0$=.9,$\alpha$=0@epoch50 |
| **AutoSparse** | 77.01 | 75.19 | 2.36 | 89.94 | 0.36× | 0.081× | $\alpha_0$=.9,$\alpha$=0@epoch45 |
| STR | 77.01 | 70.4 | 8.58 | 95.03 | 0.246× | 0.039× | dense to sparse |
| **AutoSparse** | 77.01 | 70.84 | 8.01 | 95.09 | 0.182× | 0.036× | $\alpha_0$=0.8,$\alpha$=0@epoch20 |

Table 1: ResNet50 on ImageNet: Comparing accuracy, sparsity and the FLOPS (dense 1×) for training and inference for selected sparse training methods. **TopKAST***: TopKAST with 0% backward sparsity. **TopKAST*+GA**: TopKAST* with Gradient Annealing boosts the accuracy despite having same training/inference FLOPS. For AutoSparse 79.67% sparsity, $\alpha_0$=0.75 is decayed till epoch 90 and then set to 0 (implying ∼ 80% forward and backward sparsity after epoch 90). Similarly, for AutoSparse 95.09% sparsity, $\alpha_0$=0.8 is decayed till epoch 20 and then set to 0. Standard deviation of results for AutoSparse is less than 0.06.

| Method | S% | Top1(S) | Drop% | Train F | Test F | Back(S) | comment |
|---|---|---|---|---|---|---|---|
| **AutoSparse** | 83.74 | 75.02 | 2.58 | 0.328× | 0.128× | 50 | $\alpha_0$=1.0, $s_0$=-8, w grad 50% sparse |
| TopKAST | 80 | 75 | 2.34 | 0.32× | 0.22× | 50 | fwd & in grad 80%, w grad 50% sparse |

Table 2: ResNet50 on ImageNet: AutoSparse with explicitly set sparsity for weight gradient. For AutoSparse 83.74% sparsity, $\alpha_0 = 1$ is decayed till epoch 100 resulting in identical sparsity for forward and input gradients, along with 50% sparsity for weight gradient throughout the training (involves invoking TopK method).

weights and sparsity mask parameters using explicit regularizer with the objective function. They report 0.23% drop in accuracy at 74% sparsity for ResNet50 on ImageNet. However, their much lower dense baseline, lower reported sparsity and lack of training FLOPs make it harder for a direct comparison (inference FLOPs 0.21× of baseline). LTP produces 89% sparse ResNet50 that suffers from 3.2% drop in accuracy from baseline (worse than ours). Continuous Sparsification Savarese et al. (2021) induces sparse training using soft-thresholding as a regularizer. The sparse model produced by them is used to test Lottery Ticket Hypothesis (retrained). Lack of FLOPs number makes it harder to directly compare with our method. GDP (Guo et al. (2021)) used FLOPS count as a regularizer to achieve optimal accuracy vs compute trade-off (0.4% drop in accuracy with 0.49× training FLOPS. Our method achieves slightly higher accuracy with lower FLOPS count.

**Comparison with Deterministic Pruning Methods:**

RigL Evci et al. (2021) activates/revives new weights iteratively using infrequent full gradient calculation. Such restriction of gradient flow causes more accuracy drop: 80% sparse model loses

| Method | Base | Top1(S) | Drop% | S% | Train F | Test F | comment |
|--------|------|---------|-------|-----|---------|--------|---------|
| STR | 71.95 | 68.35 | 5 | 75.28 | $0.37\times$ | $0.18\times$ | dense to sparse |
| **AutoSparse** | 71.95 | 70.1 | 2.57 | 75.1 | $0.51\times$ | $0.22\times$ | $\alpha \neq 0$ |
| STR | 71.95 | 64.83 | 9.9 | 85.8 | $0.32\times$ | $0.1\times$ | dense to sparse |
| **AutoSparse** | 71.95 | 64.87 | 9.84 | 86.36 | $0.25\times$ | $0.1\times$ | $\alpha$=0 @ epoch 20 |
| **AutoSparse** | 71.95 | 64.18 | 10.8 | 87.72 | $0.22\times$ | $0.08\times$ | $\alpha$=0 @ epoch 20 |

Table 3: MobileNetV1 on ImageNet: Comparing accuracy, sparsity and FLOPS (dense $1\times$) for training and inference for selected sparse training methods. AutoSparse achieves significantly higher accuracy for comparable sparsity. For AutoSparse 75.1% sparsity, $\alpha = 0.4$ is decayed till epoch 100, resulting in identical sparsity for forward and input gradient computation. For AutoSparse (87.72%,86.36%) sparsity, ($\alpha = 0.6, \alpha = 0.8$) decayed till epoch 20 and then set to 0, which implies identical forward and backward sparsity after epoch 20.

2.86% top-1 accuracy from dense baseline (table 1). ITOP Liu et al. (2021) maintains a fixed amount of sparsity throughout training. They use gradient based weight regrowth used in RigL and SNFS Dettmers & Zettlemoyer (2019). Applying ITOP in RigL grow stage, i.e., exploring new weights based on (full) gradient information in Rigl grow stage, RigL+ITOP achieves accuracy 75.84 for sparsity budget 80% ∼ 1.25% off from baseline. OptG Zhang et al. (2022) learns both weights and a pruning supermask in a gradient driven manner where pruned weights also receives loss gradient. AutoSparse is better than their accuracy at 90% sparsity budget, despite achieving significantly higher average sparsity. PowerPropagation Schwarz et al. (2021) transforms weight as $w = v|v|^{\alpha-1}$, s.t. it creates a heavy-tailed distribution of trained weights that are more amenable to pruning. PP, when applied on TopKAST, i.e., TopKAST + PP, improves on TopKAST and achieves 0.73% drop in accuracy for 80% sparsity budget and 2% drop in accuracy for 90% sparsity budget. Gradient Annealing, when applied with TopKAST method (without additional regularization) improves these accuracies while maintaining similar training and inference FLOPs. SWAT-U sparsifies models in the forward pass and activation in the backpass to reduce the training FLOPs. For models with ReLU activation, we typically have 50% default sparsity. This would lead to up to $2\times$ further reduction of training FLOPS for AutoSparse. However, in our analysis we do not count such inherent sparsity of activations. Finally, MEST (SOTA for fixed-budget sparse training) achieves comparable accuracy for 80% sparse Resnet50, however, using 20% more training FLOPS and 45% more inference FLOPS (as their sparse model is not FLOPS-efficient).

## 5.3 AutoSparse for Language Models

We applied our AutoSparse on Transformer model using exponential decay for $\alpha = 0.4$. This annealing schedule resulted in a better trade-off between accuracy and sparsity. Also, we set $\alpha = 0$ after 10 epochs to take advantage of more sparse compute in backward pass.

AutoSparse achieves a BLEU score of 27.57 with overall sparsity 60.99% which incurs an accuracy drop of 0.93% from dense baseline BLEU score 27.83.

## 6 Conclusion

We propose Gradient Annealing, a novel technique that decays the gradients of pruned out weights in a non-linear manner. This potentially eliminates the need for explicit regularizer to induce sparsity in model training. This provides us with a tool to develop an automated sparse training method to explore the inherent model sparsity without the knowledge of an appropriate sparsity budget. AutoSparse helps us to achieve state-of-the-art sparsity/accuracy trade-off for two different type of CNN models: ResNet50 and MobiletNetV1 on ImageNet dataset. Further, we demonstrate the applicability of our method on Transformer models for language translation task. Interestingly, among methods producing comparable sparsity, AutoSparse learns a non-linear sparsity distribution that is the most FLOPs-efficient.

**Declaration**: The authors read and adhered to the ICLR Code of Ethics and ICLR Code of Conduct.

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
