# OpenReview forum: "AutoSparse: Towards Automated Sparse Training"
_ICLR.cc/2023/Conference — Submitted to ICLR 2023_

### Official Review · Reviewer_EMVQ · 2022-10-24

**Confidence:** 4
**Correctness:** 3
**Technical Novelty And Significance:** 3
**Empirical Novelty And Significance:** 2
**Recommendation:** 5

**Clarity, Quality, Novelty And Reproducibility:**

Clarity & reproducibility:
- Some hyperparameters are introduced but not introduced elsewhere, for example, the ``s_init'' on page 7 line 1.
- Some details are missing and hard to infer from the paper. What is the exact form of $g(\cdot)$ in Section 3? and $T_l$ is only defined but never used.
- (Cont'd) The gradient annealing technique is only applied for a few epochs. Is the value of $\alpha$ decayed to $0$ in, say 10 epochs, or just cut off after 10 epochs?
- If decaying to $0$ in 10 epochs, then what is the value of $L_0$, $L_1$, etc. that helps control the sparsity level? Also, I was wondering if one wants to achieve $70\%$ sparsity, could the proposed method achieve this goal?
- Reporting FLOPs is helpful and fair, but it would be better to see the reduction in wall-clock time if any. Structured pruning methods may save fewer FLOPs but can lead to a reduction in training time.

Quality & Novelty:
- The division of sparse training methods missed some pruning methods that are (1) not using fixed sparsity budget and (2) not using learnable threshold pruning methods, such as CS [1], supermasks [2,3], GDP [4]. It would be helpful to discuss these works (and maybe compare them).
- I am not fully convinced of the practical benefits of gradient annealing due to (1) the gradient annealing is only applied on a small subset of weights (B\A); (2) the technique is only applied for a short time; (3) the advantage of the non-linear scheme over linear decay is not justified in terms of the performance of the trained model. To be more specific, using layerwise learned threshold with a constant 0, constant 1, or linearly decay $\alpha$ may also achieve good performance. The same for the experiments combining TopKAST with GA: using a constant or linear decay $\alpha$ may also improve the performance.
- Threshold-based methods are commonly used for pruning, for example K Azarian et al. [5] proposed a very similar formulation. The authors should consider cite this work and compare with them.

[1] Winning the Lottery with Continuous Sparsification

[2] Deconstructing lottery tickets: Zeros, signs, and the supermask

[3] Signing The Supermask: Keep, Hide, Invert

[4] GDP: Stabilized Neural Network Pruning via Gates with Differentiable Polarization

[5] Learned Threshold Pruning

**Strength And Weaknesses:**

Strengths:

- Provided intuitive examples and explanations (Page 3-4)
- Substantial amount of experiments on large-scale datasets (ImageNet and WMT).

Weakness:

- Some details are hard to follow.
- Evaluation is not sufficient to justify the effectiveness of the proposed techniques

**Summary Of The Paper:**

This paper first introduces a technique called Gradient Annealing, which utilizes a similar concept of the straight-through estimator to enable weight updates on the pruned weights. Specifically, the gradient on the pruned weights will be shrunk by a factor, which is decaying over time. They argue that using a non-linear annealing scheme can help stabilize the sparse budgets. The authors further propose a sparse training algorithm called AutoSparse, that involves a layer-wise learnable threshold for pruning, and also applies the Gradient Annealing technique on a subset of parameters and can enhance efficiency as claimed by the authors. Experiments show performance gain on image classification and NLP tasks.


**Summary Of The Review:**

Overall I appreciate the authors' work, and it could be the community's interest and inspire future research. However, there are still several things that I believe are necessary to address. Therefore I give a marginal score, but I am also open to changing my score based on the authors' responses.

---

### Official Review · Reviewer_djDX · 2022-10-27

**Confidence:** 5
**Correctness:** 3
**Technical Novelty And Significance:** 2
**Empirical Novelty And Significance:** 2
**Recommendation:** 6

**Clarity, Quality, Novelty And Reproducibility:**

The code snippets are provided in the appendix along with the required params for each of adoption.

The rest of the points are covered in the earlier section.

**Strength And Weaknesses:**

I will go sequentially for both strengths and weaknesses.

Strengths:
1) The paper is well motivated and understands the one major problem plaguing learnable sparsity methods and exposits well.
2) The related work is well-placed.
3) The explanation and exposition of the GA method, design choices, and analysis are done well.
4) The experiments are thorough and in line with the baseline papers like STR.
5) The experiments with training and inference metrics are appreciated.
6) Backward sparsity results showcase the generality of GA

Weakness:
1) The writing is not clear and would definitely benefit from revision during the course of the discussion. This also includes the aesthetics and the issues with citation style (whatever is used is not the default for ICLR 2023) -- The second aspect has not affected my perception of the paper, but I would recommend a revision to fix them.
2) While the proposed solution of GA is interesting and states that it alleviates some of the hparam issues of learnable sparsity, its design choices often are indicative of the potential search through human function design. For example, the non-linear decay comes from the observation of not letting sparsity hit 100% soon for STR when the s_init is small in magnitude. While I like that the design helps with better scalability and applicability, it feels like each of the aspects is being handled specifically -- not a major weakness but something that struck me.
3) AutoSparse is a generalized formulation of STR, however, when comparing, AutoSparse was only compared to STR. I think it makes a lot of sense to compare AustoSparse to STR + STE (by setting $\alpha=1$) -- This experiment would show if GA is the major factor for the gains or if is it just the dead gradients through ReLU -- this is a major concern.
4) In Table 2, why don't we have an apples to apples comparison with the same sparsity for TopKAST as it is deterministic -- I would love to see the accuracy of 85% sparse topKAST solution.
5) While the application to LM is interesting, it serves little purpose in helping us understand without any baseline -- Please add a strong baselines like STR on AutoSparse for this -- I understand STR might not have done this, but it is the duty of the authors to apply an existing technique and its obvious variants as baselines for a strong paper.
6) Lastly, I also do not agree with the claim that the top-k sorting in deterministic pruning is a compute-intensive step. -- any thoughts?

I am open to discussion when the rebuttal and revisions come with answers to my mentioned weaknesses. I am looking forward to it and potentially changing my score.

**Summary Of The Paper:**

The paper introduces Gradient Annealing (GA) as a key component for learnable sparsity training to further improve training and inference efficiency alongside accuracy. The authors propose AutoSparse, a training sparse training algorithm that combines GA with the generalized formulation of STR (Kusupati et al., ICML 2020). The paper argues that GA is a crucial component and improves accuracies further compared to vanilla STR formulation, at the same time also improving accuracy for deterministic pruning methods like TopKAST.

The paper also provides the choices for the GA hyperparameter ($\alpha$) alongside an analysis of why having a GA-style non-linear decay of gradients of potentially inactive weights helps in having higher accuracy than an STR + STE variant or STR with no GA.

The paper also supports the method with extensive experimentation on ImageNet with MobileNet and ResNet50 compared against various baselines across efficiency and accuracy metrics both for STR style methods and deterministic methods. Auto Sparse was also extended to an LM task.

Note that the brevity of the review should not be taken as a negative aspect, as the paper's point is clear and direct given the similarity to STR in the formulation.

**Summary Of The Review:**

While the idea of GA alleviates some of the issues in learnable sparsity, it brings in a bit of design complication -- which is fine. However, the bigger concerns are with comparisons to some missing baselines as well as the writing. I think the paper has potential and with a strong revision is worth publishing.

----
After rebuttal, I think the paper might be a good  addition to the iclr community. However, it still has issues that need to be addressed as pointed by the other reviewers as well.

---

### Official Review · Reviewer_G8BR · 2022-10-27

**Confidence:** 5
**Correctness:** 3
**Technical Novelty And Significance:** 1
**Empirical Novelty And Significance:** 1
**Recommendation:** 3

**Clarity, Quality, Novelty And Reproducibility:**

To sum up, the clarity and quality of this paper need to be improved. The novelty is lacking since the proposed method is incremental. Please refer to strengths and weaknesses for more information.

**Strength And Weaknesses:**

Strengths

1. This paper investigates different types of non-linear annealing methods and did a thorough analysis

2. The author of the paper not only performs experiments on CV models but also on Language models.


Weaknesses

1. The overall contribution is incremental to existing approaches. Investigating and adding different annealing methods to sparse training is not a major contribution. Additionally, I don’t see a very strong motivation for using this GA method. It is more like a training “trick” of dynamic sparse training.

2. The accuracy improvements are not significant. Many existing works on sparse training are not compared, such as MEST (NeurIPS 2021) or GradMax (ICLR 2022).

3. Writing is not clear. For example, in the introduction, the author first claims deterministic pruning sets desired fixed sparsity for each layer before training, then at the deterministic pruning subsection, they claim method [26] (DSR) is belong to deterministic pruning. However, DSR changes sparsity distribution during training, which is not what the author claimed in the earlier part of this paper. So I think some of the writing is not rigorous.

4. It is not true that non-uniform sparsity distribution “did not show any improvement” to uniform distribution. RigL (ICML 2020) and GaP (ICLR 2022, Effective Model Sparsification by Scheduled Grow-and-Prune Methods) are all showing non-uniform sparse distribution achieves lot more better accuracy. So it is not rigorous to have that strong claim in this paper without any experimental data to prove such claim.

5. The overall writing and paper organization is not good. The introduction part is not entirely clear to me, and the structure is confusing. Some part of the introduction feels like a related work survey.

6. Very limited experimental results in this paper. The author didn’t show any data for motivation, and no ablation study showing why the proposed method is effective. No analytical or empirical experiments are performed to convince the reader on why the results is outperforming others.


**Summary Of The Paper:**

This paper proposes gradient annealing method in dynamic sparse training to improve the performance. Based on gradient annealing, the author of the paper proposes AutoSparse algorithm that uses a learnable threshold to find better sparse topology and sparsity distribution during DST. Multiple experiments are conducted on ImageNet.

**Summary Of The Review:**

 I think this paper needs major revision, both on the technical contribution and writing. My suggestion is reject.

---

### Official Review · Reviewer_5Hu8 · 2022-10-27

**Confidence:** 5
**Clarity, Quality, Novelty And Reproducibility:** See the strength and weaknesses.
**Correctness:** 2
**Technical Novelty And Significance:** 2
**Empirical Novelty And Significance:** 1
**Recommendation:** 3

**Strength And Weaknesses:**

Strength:
1.	This paper is well written and easy to read.
2.	The experiments show that the proposed method can find a sparse subnetwork during training.

Weaknesses:

1. Some overclaim statements. In page 4, the authors claim that “minimize sparsification overhead” and “optimal trade-off”. How to prove these optima, as the proposed method is heuristic?

2. This paper is not well motivated. In the abstract, the authors say that the proposed method eliminates the need for sparsity inducing regularizer. I don’t think this is an advantage. Moreover, in the proposed method, it seems that it is difficult to control the sparsity of the finally trained neural networks, which is very important in real applications.

3. The authors propose some tricks to sparisfy the computation in training, such as in sparsifying the weight gradient. It is unclear whether these tricks would produce bad impacts in training as no theoretical analysis is provided in this paper.

4. The authors claim that the proposed method can reduce the FLOPS in training. Since the structure of sparse computation in this method is complicated, how to calculate/count the computational savings in FLOPs? The authors are recommended to release the code to  show this counting procedure.


**Summary Of The Paper:**

In this paper, the authors propose a new sparse neural network training method. The key component in the proposed method is called Gradient Annealing (GA). GA can automatically find a sparse subnetwork in the end of training. The authors also propose some tricks to sparsify the computation in training. A series of experiments are used to evaluate the performance of the proposed method.

**Summary Of The Review:**

1. The proposed method seems tricky and not elegant. It is unclear whether these tricks would produce negative impacts in training.

2. This paper is not well motivated.

3. It is difficult to control the final sparsity in the proposed method, which is very important in real applications.

4. Some details, such as how to calculate the computational saving, are missing.

---

### Official Review · Reviewer_7rY9 · 2022-10-31

**Confidence:** 3
**Correctness:** 2
**Technical Novelty And Significance:** 2
**Empirical Novelty And Significance:** 2
**Recommendation:** 5

**Clarity, Quality, Novelty And Reproducibility:**

Clarity: The paper writing style is a bit verbose and perhaps misleading in some places. In needs a careful revision in order to make sharp clear the actual paper contributions, the underlying mechanisms/behaviour behind the proposed methodology, while reflecting accurately the literature. The paper would benefit also from a discussion in conclusion about the limitations of the work.

Quality: Can be improved considerably.

Originality: Likely, small but ok.

Reproducibility: I see that the code of the proposed method is printed in an appendix. It would be much more beneficial for the reader to provide a full prototype open-source code which runs at a click.


**Strength And Weaknesses:**

Strength:
* Interesting idea with a small level of novelty
* The proposed method seems to be able to slightly improve the performance on top of the baselines.
* Experiments performed on large datasets and models

Weaknesses:
* The categorization of sparse training methods from the first page and the related work is shallow and misleading. For instance, “deterministic pruning” takes the reader to the idea of a deterministic process, while in reality just the sparsity level may be fixed (or some rules of pruning a part of the connections, etc.), but the output itself (e.g., the sparse connectivity) is a result of (in most cases) a random process (e.g, random pruning, random sparse initialization, stochastic gradient descent, etc). This needs serious adjustment (including of a number of statements in the paper) to reflect better the state-of-the-art and to clarify sharply the paper contributions on top of the existing work. One could start from latest survey papers such as [1]
* The proposed method has a small level of novelty. Some of the main paper claims are perhaps too strong.
* The improvement achieved by the proposed method over the baselines seems marginal. The statistical significance of the results has to be studied in order to boost the paper quality. If the large datasets/models hinder this operation then smaller datasets and other type of layers (e.g., fully-connected) could contribute in offering a more comprehensive understanding of the proposed method behaviors.
* Minor: TopKAST doesn't achieve anymore the state of the art performance. For instance, the results reported in [2] seems higher than the results achieved in this paper. A direct comparison or a study to see if the proposed method can improve also MEST is necessary. Please note that this may lead to claims re-adjustments.
* Minor: Some parts of the used terminology can bring confusion. It has to be made more rigorous. For instance, GA is an acronym typically used for Genetic Algorithms.

Non-exhaustive list of missing references:

[1] Torsten Hoefler, Dan Alistarh, Tal Ben-Nun, Nikoli Dryden, Alexandra Peste, Sparsity in Deep Learning: Pruning and growth for efficient inference and training in neural networks.
Authors, JMLR 2021, https://www.jmlr.org/papers/volume22/21-0366/21-0366.pdf

[2] Geng Yuan, Xiaolong Ma, Wei Niu, Zhengang Li, Zhenglun Kong, Ning Liu, Yifan Gong, Zheng Zhan, Chaoyang He, Qing Jin, Siyue Wang, Minghai Qin, Bin Ren, Yanzhi Wang, Sijia Liu, Xue Lin, MEST: Accurate and Fast Memory-Economic Sparse Training Framework on the Edge, NeurIPS 2021, https://arxiv.org/abs/2110.14032




**Summary Of The Paper:**

The paper proposes Gradient Annealing (GA) and AutoSparse, two complementary approaches to achieve a towards optimal sparsity-accuracy trade-off during the training of sparse neural networks.

**Summary Of The Review:**

Interesting work, but it seems to be a bit immature and not ready yet for publication.

---

### Official Review · Reviewer_Xqxh · 2022-11-01

**Confidence:** 2
**Correctness:** 3
**Technical Novelty And Significance:** 3
**Empirical Novelty And Significance:** 3
**Recommendation:** 6

**Clarity, Quality, Novelty And Reproducibility:**

- The quality of the writing is good, though there are some typos:

  - In Section 5.1, "largescale" -> "large-scale".

  - In Section 5.1, "warm up 4000,," -> "warm up 4000,"



- The technical contributions seem to be novel, though I am not an expert in this field.

- This paper provides the demo code in the appendix, so it appears to me that this work can be reproduced based on the current version.

**Strength And Weaknesses:**

### Strengths

- This paper is well-written and easy to follow.

- The technical contributions,  i.e., GA, a gradient-based non-linear method, for addressing the trade-off for model sparsity and accuracy, are novel.

- The experimental evaluation is solid, including baseline comparisons in the main paper and ablation studies in the appendix.



### Weaknesses

- Vision Transformers (ViT) is a popular backbone network in recent years. The empirical evaluations shall conduct baseline comparisons by using ViT on ImageNet.



[1] An Image is Worth 16x16 Words: Transformers for Image Recognition at Scale, ICLR 2021.

**Summary Of The Paper:**

This paper focuses on sparse training, which aims to reduce the computational overhead of deep neural networks. Specifically, this work proposes a non-linear gradient-based method, namely, Gradient Annealing (GA), to address the trade-off between model sparsity and accuracy. Meanwhile, this paper combines one latest sparse training method with GA, arriving at a unified training algorithm, i.e., AutoSparse. Extensive experimental results demonstrate that the proposed method could achieve the state-of-the-art model sparsity of 80% on ResNet and of 75% on MobileNetV1. Besides, GA outperforms TopKAST+PP by 0.3% in terms of classification accuracy on ImageNet.

**Summary Of The Review:**

This paper proposes Gradient Annealing (GA) to address the trade-off between model sparsity and accuracy, and then, combine GA with the recent state-of-the-art method, arriving at a unified training algorithm called AutoSparse. The technical contributions seem to be novel. Meanwhile, this paper is well-written and the proposed method is reproducible. Based on the above consideration, I recommend accepting this paper. But, since I am not an expert in this field, I am open to changing my score, based on the authors' responses and other reviewers' comments.

---

### Decision · Program_Chairs · 2023-01-20

**Decision:**

Reject

**Justification For Why Not Higher Score:**

See above

**Justification For Why Not Lower Score:**

N/A

**Metareview: Summary, Strengths And Weaknesses:**

This paper proposes gradient annealing method in dynamic sparse training, and their "AutoSparse" algorithm uses a learnable threshold to find better sparse topology and sparsity distribution during DST. Multiple experiments are conducted on ImageNet.

Several weaknesses are identified by reviewers, and the rebuttal doesn't seem to clarify those concerns:
- Weak comparison and positioning w.r.t the vast sparse training literature.  The authors used some self-crafted, somehow misleading terminologies. It's not a good idea to skip prior art discussion only because it " requires lengthy discussion that is hard to fit in few pages. "
- The novelty is lacking and overall incremental (various gradient annealing for sparse training methods)
- The improvement achieved by the proposed method over the baselines seems marginal. The practical benefit of gradient annealing on a small subset of weights is also questionable.

The reviewers in general lack enthusiasm about this paper. Reading all comments, AC doesn't find the ground to support acceptance.